

# Climate change impacts on groundwater simulated using the AquiFR modelling platform

Alexis Jeantet[1,2], Jean-Pierre Vergnes[3], Simon Munier[2], and Florence Habets[1]

[1]Laboratoire de Géologie, CNRS UMR 8538, École Normale Supérieure, PSL University, IPSL, Paris, France
[2]CNRM, Université de Toulouse, Météo-France, CNRS UMR 3589, Toulouse, France
[3]Water, Environment, Processes and Analyses Division, BRGM – French Geological Survey, Orléans, France

**Correspondence:** Alexis Jeantet (alexis.jeantet@meteo.fr)

**Abstract.** In the context of increasing water stress and climate change, the assessment of changes in groundwater resources is a major challenge for water decision-makers. As part of the EXPLORE2 project, the aim of this study is to estimate changes in groundwater levels over France during the $21^{st}$ century. We used the hydrogeological modelling platform AquiFR together with 36 regional climate projections from Eurocordex (CMIP5) from three Representative Concentration Pathways (RCPs), bias-corrected according to a state-of-the-art method: RCP2.6, RCP4.5 and RCP8.5. The future evolution of groundwater is assessed using the standardized piezometric level index, a normalized indicator that provides return periods based on the distribution value over a reference period, here 1976-2004. We found significant scatters, especially between regional climate models. Overall, a rise in groundwater levels, affecting most of the study area, is the dominant signal, especially in northern France. This result is in contrast to previous studies in this area. Under RCP8.5 (highest greenhouse gas emissions scenario), the evolution of the occurrence of current 10-year return period events shows a significant increase in the risk of high groundwater levels mostly on the northern part of France, together with an increase in the 10-year low groundwater levels mostly observed in South of France, which highlights a North-South differentiation. The increase in high and low flow events is quite common in surface hydrology, but is less common for groundwater, which has a longer residence time. In order to better reflect the uncertainties, 4 story lines based on the RCP8.5 scenario have been selected to be representative of possible futures that can illustrate the impacts of worst-case scenarios and help decision-makers to adopt sustainable groundwater management policies.

## 1 Introduction

In France, groundwater supports 70% of drinking water, for both agricultural purposes, accounting for one-third of agricultural withdrawals (Pasquier, 2017), and supply of drinking water to households (Maréchal and Rouillard, 2020). Therefore, understanding and anticipating future groundwater temporal evolution is crucial for stakeholders and decision-makers involved in the sustainable management of water resources, especially because this evolution depends on both human activities and climate (Baulon et al., 2020; Guillaumot et al., 2024). Furthermore, the Intergovernmental Panel on Climate Change (IPCC) states that a large proportion of hydrological systems (e.g., surface rivers, groundwater tables, ...) are already being affected on a global scale and will be further affected in the coming decades by climate change (IPCC, 2014, 2022). Among the projected changes figures intensification of extreme events in terms of floods and droughts, in both frequency and intensity (Kundzewicz et al.,



2014; Prudhomme et al., 2014; Garner et al., 2015), potentially affecting groundwater management.

Several projects have provided a first global assessment of the consequences of climate change on French hydrology, e.g., the EXPLORE 2070 project (Chauveau et al., 2013) using climate projections provided by the Coupled-Model Intercomparison Project 3 (CMIP3) global exercise, or more recent projects using climate projections from the CMIP5 global exercise (Taylor et al., 2012) regionalized over France by Dayon et al. (2018), to assess the future of groundwater on specific watersheds (Boé

et al., 2018; Habets et al., 2021). These studies projected a widespread drop in groundwater levels induced by a decrease in recharge of between 10 and 25%, with spatial variations (Stollsteiner, 2012; Habets et al., 2013). Two areas were more severely affected: the Loire basin, with a decrease in recharge of between 25 and 30% over half of its surface; and especially South-West France, with decreases of between 30 and 50%. Completed with surface hydrology assessment, the EXPLORE 2070 project allowed to better measure the magnitude of the challenge of defining national adaptation or attenuation strategies, regarding

numerous sectors of activity, e.g., energy and agriculture (Carroget et al., 2017).

However, more recent studies suggest that the impacts of climate change on French hydrological systems, including groundwater, are more contrasted, especially in northern France, (Costantini et al., 2023; Vergnes et al., 2023), highlighting the need of a new general assessment of climate change impacts over the French hydrology. Following the IPCC recommendations (IPCC, 2022), the EXPLORE2 project (Marson et al., 2024; Sauquet et al., 2024) aims to update the conclusions of the EXPLORE

2070 project for this purpose. Regarding groundwater, this project is an opportunity to integrate larger study areas and more aquifers of interest.

In such a context, the study of climate change on hydrological systems requires the use of a hydroclimatic modelling chain, ranging from updated greenhouse gas emission scenarios to hydrological indicators, and established on a multi-model ensemble approach (Sansom et al., 2013). The principle is to use several radiative forcing scenarios, climate models and hydro(geo)logical

models to quantify the uncertainty associated with the projections (Knutti et al., 2013; Rajib et al., 2014). Indeed, each climate model differs in elements such as precipitation distribution (Lafaysse et al., 2014; Shepherd, 2019) which necessarily leads to different projections at the end of the modelling chain. Consequently, using multiple models at each step is strongly recommended (Hingray and Saïd, 2014; Vidal et al., 2016; Hingray et al., 2019). Eventually, the challenge for modellers is to assess the uncertainties inherent to the amount of obtained results, for example by identifying converging patterns or, conversely, the

absence of a robust trend from all projections. (Kennel et al., 2016).

In order to provide consistent information on groundwater, the use of models providing realistic representations of aquifer systems is required (Vergnes et al., 2020). Therefore, the hydrogeological part of the modelling chain is often composed of spatially distributed hydrogeological regional models built over specific studies areas at the regional scale (Stollsteiner, 2012; Højberg et al., 2013; Maxwell et al., 2015). In this respect, the hydrogeological platform AquiFR (Vergnes et al., 2020) is a rel-

evant candidate as the platform integrates four distributed hydrogeological modelling software packages with several regional applications, in order to account for the heterogeneity of French aquifers. Coupled with the SURFEX land surface model to provide recharge over its entire domain (Masson et al., 2013; Le Moigne et al., 2020), the AquiFR system is already operational for historical reanalysis, real-time monitoring and seasonal forecast of groundwater, especially for the assessment of dry events to help decision-makers (Habets et al., 2021). As part of a study on the impact of climate change on French groundwater,





CMIP5 global climate projections has already been used by AquiFR (Habets et al., 2021). Since, projections have been updated and a new assessment is required, which highlights the interest of the present paper.

As part of the EXPLORE2 project, the aim of this study is to assess the impact of climate change projected by regional climate models on several French aquifers of interest. For this purpose, the hydrogeological modelling platform AquiFR is used to simulate hydrogeological projections using an multi-model ensemble approach based on 36 climate projections that used three future socio-economic scenarios. These projections are a subset of the Euro-CORDEX project (Jacob et al., 2020) resulting from the CMIP5 exercise (IPCC, 2014), downscaled with a method combining weather typing and quantile mapping approaches, that are made available on a public database (DRIAS, Soubeyroux et al. (2021)).

## 2    Material and Methods

### 2.1    Regionalized climate projections from EXPLORE2

The climate projections (CPs) used to force the AquiFR platform mainly come from the DRIAS-2020 database (DRIAS 2021; **http://www.drias-climat.fr/**, last access the 25th of October, 2024) that provides downscaled and unbiased climate projections on the French domain. It is based on a subset of projections from the Euro-CORDEX project (CooRdinated Downscaling EXperiment – European Domain; Jacob et al. (2020)) of the international Coupled Intercomparison Model Project Phase 5 (CMIP5; IPCC (2014)). As part of the implementation of the EXPLORE2 database, the original DRIAS-2020 database has been extended to include the latest EURO-CORDEX simulations with evolving aerosol forcing (Robin et al., 2023) leading to the EXPLORE2-Climat 2022 database.

To obtain future climate variables, General Circulations Models (GCMs; Phillips (1956); Randall (2000)) are forced by Radiative Concentration Pathways (RCPs), greenhouse gas emission scenarios following specific socio-economical trajectories through the $21^{st}$ century. However, GCM outputs have a coarse spatial resolution around 2.5°, i.e., between 100 and 200 km$^2$, which is insufficient to adequately represent local meteorological phenomenons and extreme events, and thus prevents their direct use as climate forcing in hydrological models for studies at regional or local scales. Thus prior to use a statistical downscaling method. A dynamic downscaling is made by using RCMs (Regional Climate Models), which provide detailed estimates of meteorological parameters (e.g., temperature, precipitation, humidity, wind, solar radiation, ...) at an hourly time step for regional applications using a dynamic downscaling approach (Liang et al., 2008; Tapiador et al., 2020), at a spatial from 10 to 20 km$^2$.

Then, the ADjustment to MOuNTain regions (ADAMONT; Verfaillie et al. (2017)) was applied to the RCM outputs to both improve spatial resolution and correct bias according to an historical meteorological reference by a two-steps method:

1. Weather regime computation (Michelangeli et al., 2009; Driouech et al., 2010): each day from both the RCMs and the meteorological reference are clustered into four weather regimes, for each of the four seasons: (1) December, January, February (DJF); (2) March, April, May (MAM); (3) June, July, August (JJA); (4) September, October, November (SON).



**Table 1.** Availability of climate projections. The numbers (2.6, 4.5, and 8.5) refer to the RCPs used by the GCM (rows)/RCM (columns) couples. "-" indicates missing data

| | ALADIN 63 | RACMO 22E | HadREM3 -GA7-05 | RCA4 | HIR HAM5 | WRF 381P | CCLM4 -8-17 | RegCM4 -6 | REMO 2009 | REMO 2015 |
|---|---|---|---|---|---|---|---|---|---|---|
| CNRM -CM5 | 2.6<br>4.5<br>8.5 | - | 8.5 | - | - | - | - | - | - | - |
| EC- EARTH | - | 2.6<br>4.5<br>8.5 | 2.6<br>8.5 | 2.6<br>4.5<br>8.5 | - | - | - | - | - | - |
| IPSL-CM5 -MR | - | - | - | 4.5<br>8.5 | 8.5 | - | - | - | - | - |
| HadGEM2 -ES | 8.5 | - | 2.6<br>8.5 | - | - | - | 4.5<br>8.5 | 2.6<br>8.5 | - | - |
| MPI-ESM -LR | - | - | - | - | - | - | 2.6<br>4.5<br>8.5 | 2.6<br>8.5 | 2.6<br>4.5<br>8.5 | - |
| NorESM1-M | - | - | - | - | 4.5<br>8.5 | 8.5 | - | - | - | 2.6<br>4.5<br>8.5 |

Then, quantile distributions of each climate variables from both the RCMs and the meteorological reference are computed for the 16 weather regimes/seasons couples (Verfaillie et al., 2017);

2. Quantile mapping (Maurer et al., 2010; Navarro-Racines et al., 2020; Potter et al., 2020): for each couple, quantile mapping is applied on the quantile distributions of RCMs climate variables being adjusted according to the meteorological reference.

The historical meteorological reference used is the SAFRAN reanalysis (Vidal et al., 2010), providing "pseudo-observed" data, useful for bias correction all over France.

At the end of the procedure, the CPs simulated by the GCM/RCM couples provide downscaled climate variables at hourly time step and on a regular grid of 8 km × 8 km. The total set gathers 6 GCMs coupled with 9 RCMs, and forced by 3 RCPs : (1) RCP2.6, a stringent mitigation scenario assuming an efficient environmental international policy (van Vuuren et al., 2007); (2) RCP4.5, an intermediate stabilization scenario (Thomson et al., 2011); (3) RCP8.5, a very high green house gaz emission scenario without efficient environmental international policy (Riahi et al., 2011). The EXPLORE2 project uses 17 GCM/RCM pairs forced by 1 to 3 RCPs, for a total of 36 simulations (Table 1), all downscaled and bias corrected by ADAMONT. Following the recommendations of the EXPLORE2 project, the analyses of CPs were done on four time periods regrouping 1 historical period and 3 future horizons for a total period ranging from 1976 to 2099, considering that an hydrological year is set from



august $01^{st}$ to july $31^{th}$ (Table 2). Furthermore, four storylines, i.e., CPs based on contrasted trends, were selected from the

**Table 2.** Temporal splitting of the climate projections from the EXPLORE2 project

| Historical period | Short-term horizon | Intermediate horizon | Long-term horizon |
|:---:|:---:|:---:|:---:|
| 1976-2005 | 2021-2050 | 2041-2070 | 2071-2099 |

GCM/RCM couple in the EXPLORE2 project using the RCP8.5, in order to focus on four contrasted possible futures(Shepherd et al., 2018; Shepherd, 2019) :(Marson et al., 2024):

1. Orange storyline (EC-EARTH/HadREM3-GA7-05 couple): climate affected by a strong warming and very dry in the summer;

2. Green storyline (HadGEM2-ES/ALADIN63 couple): serious warming and increased rainfall;

3. Purple storyline (HadGEM2-ES/CCLM4-8-17 couple): serious warming and strong seasonal contrast;

4. Yellow storyline (CNRM-CM5/ALADIN63 couple): future changes remaining relatively low;

## 2.2 The AquiFR modelling platform

AquiFR is a hydro-meteorological modelling platform based on a multi-model approach to simulate french rivers and groundwater systems (Habets et al., 2021). The platform was assessed on a sixty-year periodVergnes et al. (2020), and has demonstrated good performance in simulating groundwater level, compared to observed piezometric heads from the ADES ("Accès aux Données sur les Eaux Souterraines") database (**http://www.ades.eaufrance.fr/**, last access the 29th November 2024, Chery and Cattan (2003)). Nowadays, the AquiFR platform is used in an operational context for real-time monitoring and seasonal forecasting on the main sedimentary basins and aquifers for groundwater decision-makers (Habets et al., 2021).

### 2.2.1 The SURFEX modelling platform

A specific feature of AquiFR is that the SURFEX modelling platform ("SURFace EXternalisée" in French; Masson et al. (2013)) is used to solve the water and energy balances at the surface-atmosphere interface. The groundwater recharge and surface runoff simulated by SURFEX is used to force the hydrogeological models implemented in AquiFR.
More specifically, only the ISBA ("Interactions Sol - Biosphère – Atmosphère" in French; Noilhan and Planton (1989)) surface scheme included in SURFEX to simulated the continental area is used. SURFEX allows a direct coupling with an atmospheric model, with a spatial resolution adaptable to a wide range of grid sizes, from hundreds of kilometers (e.g., when coupled to a global climate model) to a few kilometers for more local studies, for example coupled with the MODCOU model (Le Moigne et al., 2020) or the CTRIP model (CNRM version of the Total Runoff Integrating Pathways model; Decharme et al. (2019); Munier and Decharme (2022)) for hydrological studies.



The atmospheric forcing used by SURFEX varies depending on the purpose, either the SAFRAN historical reanalysis ("Système d'Analyse Fournissant des Renseignements Adaptés à la Nivologie" in French; Vidal et al. (2010)) for historical reanalysis
and real-time monitoring or seasonal forecasting (Roehrig et al., 2020).

### 2.2.2  Hydrogeological modelling software

At the time of this study, four hydrogeological modelling software packages are integrated into the AquiFR platform: the physically-based modelling softwares EauDyssée (Saleh et al., 2013) and MARTHE ("Modélisation des Aquifères avec un
maillage Rectangulaire, Transport et HydrodynamiquE" in French; Thiéry (2015)) devoted to sedimentarty aquifer system; EROS ("Ensemble de Rivières Organisées en Sous-bassins" in French; Thiéry (2018)) adapted for karst spring discharge modelling; and HS1D (1D hillslope model; Marçais et al. (2017)) for hardrock aquifers. However as EROS and HS1D only provide surface flows, these 2 models have not been taken into account in this study.

EauDyssée is an updated version of the MODCOU modelling software (Ledoux, 1980) gathering several modules. The most
important ones are the SAM module ("Simulation des Aquifères Multi-couches" in French; Ledoux et al. (1989)) for hydrogeological processes, and the RAPID module (Routing Application for Parallel computation of Discharge; David et al. (2011)) for river routing. SAM simulates multilayer aquifers using a finite difference numerical scheme to solve the groundwater diffusivity equation. SAM supposed 2D horizontal groundwater flows and vertical exchange through aquitards in order to represent both unconfined and confined aquifers (Vergnes et al., 2020). Transfer through the unsaturated zone is explicitly taken into
account with a simple approach based on a Nash cascade (Philippe et al., 2011). The RAPID module uses matrix-based version of the Muskingum routing scheme to calculate discharge simultaneously through a river network. RAPID is coupled with groundwater models, which allows exchanges with groundwater in both directions.

MARTHE is the hydrogeological modelling software program from the French Geological Survey (BRGM) which simulates coupled groundwater flows and mass transfers and rivers flows using single-layer to multilayer aquifers and hydrographic net-
works. MARTHE simulates groundwater flows by a 3-dimensional finite volume approach to solve the hydrodynamic equation based on Darcy's law and mass conservation on a regular rectangular grids, coupled to a kinetic was approach used to simulate river flows (Vergnes et al., 2020). In the AquiFR platform, EauDyssée and MARTHE are the two physically-based models used to represent sedimentary aquifers, especially due to their capacity to simulate flows to the unsaturated zone as well as the groundwater-river exchanges (Philippe et al., 2011).

More detailed information about hydrogeological modelling software are provided in Habets et al. (2021). Let us note that, in practice, the hydrogeological modelling software packages are simultaneously and synchronously fed by the recharge and drainage simulated by SURFEX and connected via a Python software. The AquiFR results are provided in a homogeneous format defined according to an irregular spatial grid with a resolution ranging from 100 m to 2000 m on a monthly time step (Habets et al., 2021).





### 2.2.3 Spatial coverage of the AquiFR domain

In this paper, the AquiFR domain is defined as the spatial extent of the 13 models simulated within the AquiFR platform, one model being defined as a pair of catchments and the hydrogeological modelling software used Figure 1 and Table 3). These models are mainly located in the large sedimentary basin in the North part of France, including Poitou-Charentes, the Loire and the Parisian basin, up to the Nord Pas-de-Calais region. Two are located further out: MARTHE Alsace, which covers an area of around 1,100 km$^2$ East of France, and MARTHE Tarn-et-Garonne, of a similar size, which is the only in the southern part of mainland France. The surface areas of the aquifer systems within the models vary from about 7000 to 66000 km$^2$.

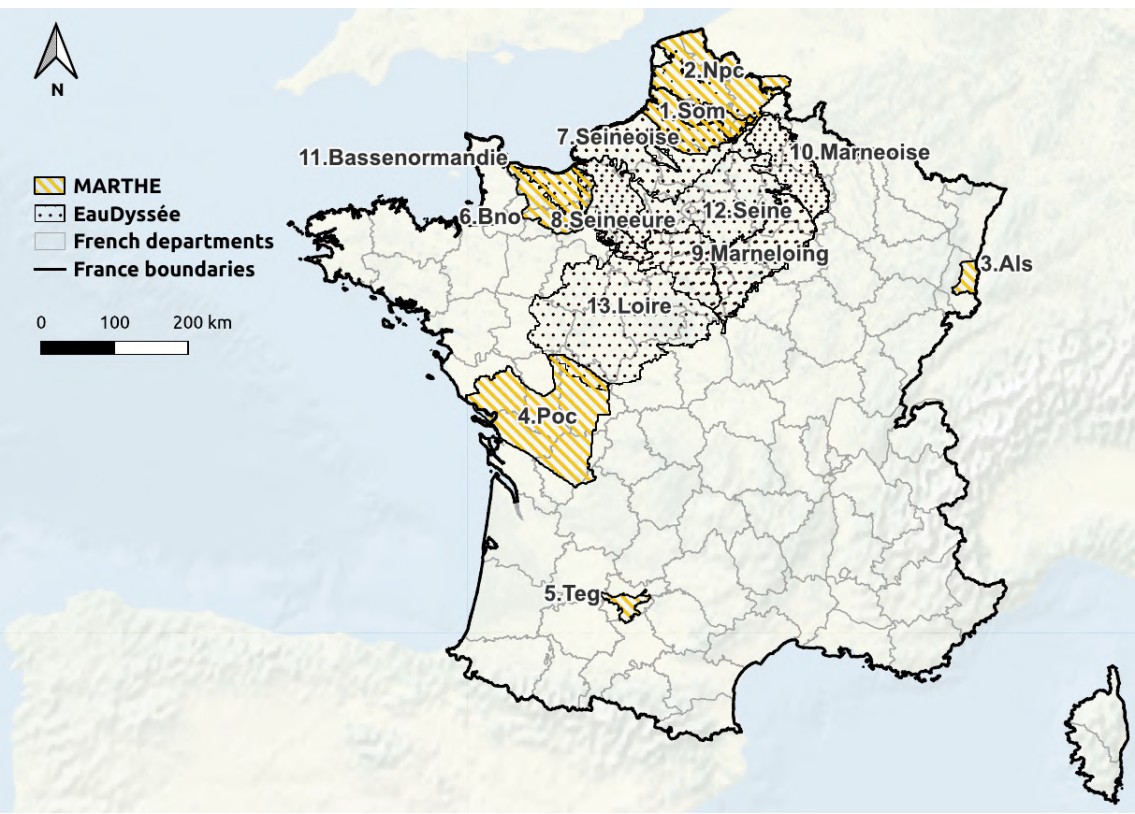

**Figure 1.** Spatial distribution of the AquiFR models. The numbers refer to the IDs of each model regrouped in Table 3.

On some basins, both EauDyssée and MARTHE are used, e.g., in Basse-Normandie. In these cases, the simulations are carried out separately using the two different models. In addition, some of the aquifer systems overlap spatially (to account for potential evolution of the groundwater edge). Therefore, part of the results overlapped when plotting, so priority is given to the most accurate models on top of the maps and to perform the corresponding analyses. This order corresponds to the numbering of the IDs presented in Table 3, from the uppermost surface layer to the deepest layer, and is applied to all spatial projection





**Table 3.** Characteristics of the 13 studied models included in the AquiFR platform.

| IDs on Figure 1 | Abbreviation | Catchment | Surface (km$^2$) | Hydrogeological model |
|---|---|---|---|---|
| 1 | Som | Somme | 7506 | MARTHE |
| 2 | Npc | Nord – Pas-de-Calais | 13332 | MARTHE |
| 3 | Als | Alsace | 1079 | MARTHE |
| 4 | Poc | Poitou-Charente | 19326 | MARTHE |
| 5 | Teg | Tarn-et-Garonne | 1139 | MARTHE |
| 6 | Bno | Basse-Normandie | 7646 | MARTHE |
| 7 | Seineoise | Seine – Oise | 15968 | EauDyssée |
| 8 | Seineeure | Seine – Eure | 9318 | EauDyssée |
| 9 | Marneloing | Marne – Loing | 10861 | EauDyssée |
| 10 | Marneoise | Marne – Oise | 7285 | EauDyssée |
| 11 | Bassenormandie | Basse-Normandie | 3626 | EauDyssée |
| 12 | Seine | Seine | 65696 | EauDyssée |
| 13 | Loire | Loire | 38746 | EauDyssée |

results in this article.

## 2.3 Method used to analyse the climate change projections

### 2.3.1 Standardized Piezometric Level Index

In the following, the evolution of the groundwater level over the AquiFR domain is analysed using the Standardized Piezo-metric Level Index (SPLI; Seguin (2015)), a normalized indicator of monthly water table level allowing a comparison with the distribution on a reference period. SPLI is very useful to evidence the evolution compared to a reference period, regardless the intrinsic hydrogeological heterogeneity of the study area. The advantage of the SPLI is that it compares time series of groundwater levels, i.e., a reference series and an assessed series, to describe the return period of an assessed event relative to the reference serie (Seguin, 2015; Vergnes et al., 2023). Therefore, the SPLI is widely used by the French Geological Survey (BRGM), as well as for seasonal forecasts with AquiFR over France.

The SPLI is based on the same approach than the well-known Standard Precipitation Index (SPI; McKee et al. (1993)). The first step consists in extracting 12 series of monthly groundwater level from the reference historical time series, separately from





January to December, from an historical series of N years (N = 30). Then, monthly cumulative distribution functions (CDFs) are extracted from the monthly groundwater levels and they are used to fit empirical distribution functions based on standard normal distributions using a non-parametric Kernell-type estimator (Figure 2-a). Eventually, the frequencies of the normal distributions are translated into SPLI values (Figure 2-b). Consequently, a groundwater level value corresponds to a SPLI value, which in turn corresponds to a frequency for the reference chronicle and therefore to a return period in that reference. In the end, each future month can be expressed by the return period of the equivalent event in the historical reference period.

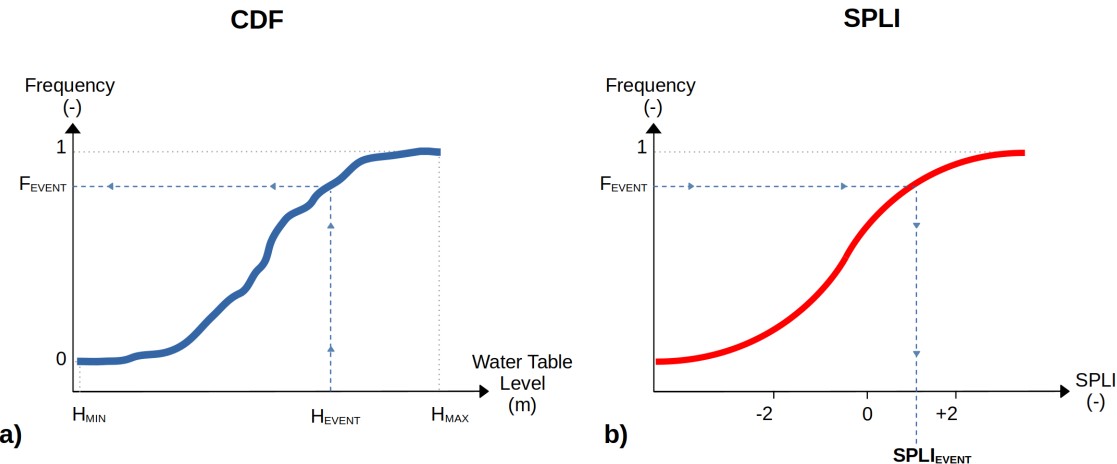

**Figure 2.** General approach for constructing the SPLI.

The SPLI values are centered on 0, which corresponds to a simulated groundwater level closed to the median of the groundwater levels in the reference series. A negative value indicates a lower level, i.e., for an event lower (and thus drier) than the median, up to -3 (corresponding to a very low water level), while positive values indicate a higher level, i.e., for a wetter event, up to +3. The range of SPLI values associated with return periods (T) is divided into seven categories (Table 4).

### 2.3.2 Groundwater evolution under climate change

In this paper, the evolution of the groundwater is first analysed regarding the spatial distribution of the median of the SPLI values from the 3 horizons of Table 2 and under the 3 RCPs. The medians of the SPLI values from each pixel of a model are extracted for each future horizon and each RCP, which illustrates the temporal evolution of the medians of the SPLI in each RCP/horizon couple. This evaluation is performed for each GCM/RCM couple in order to illustrate the uncertainty related to CPs.

A second analysis is carried out on the temporal evolution of the surface area from the AquiFR domain according to the present day return period Table 4. In practice, the SPLI values of all models are regrouped and the proportion of area affected by a specific return period is defined as the sum of the surface of each pixel dedicated to this return period, and divided by the



**Table 4.** Classification of groundwater heads (H) assessed following the SPLI values and the associated return period corresponding to the historical reference. Adapted from Vergnes et al. (2023).

| Classification | SPLI values | Return periods |
|---|---|---|
| Very low H | < -1.28 | > 10-year dry |
| Low H | [-1.28; -0.84[ | Between 5-year dry and 10-year dry |
| Relatively low H | [-0.84; -0.25[ | Between 2.5-year dry and 5-year dry |
| Around the normal state | [-0.25; +0.25] | Between 2.5-year dry and 2.5-year wet |
| Relatively high H | ]+0.25; +0.84] | Between 2.5-year wet and 5-year wet |
| High H | ]+0.84; +1.28] | Between 5-year wet and 10-year wet |
| Very high H | > +1.28 | > 10-year wet |

total surface of the AquiFR domain. This procedure is performed on the results from all the the projections. For graphical analyses, a 5-year temporal rolling median SPLI is calculated per RCP with an envelope representing the difference between the min and max values in order to compare future trends between each RCP. Because the year 2005 is characterized by the transition between the historical and the future periods into the simulations provided by the CPs, the period from 2003 to 2007 is excluded from this analysis.

The significance of the trends observed between the reference period and the horizons of the 3 RCPs is assessed using two statistical tests:

- The Kolmogorov–Smirnov test (Birnbaum and Tingey, 1951) to evaluate the significance of the difference between historical and future trends of SPLI values from 2021 to 2099 ;

- The Mann-Kendall test (Mann and Whitney, 1947) on the future SPLI values to assess if the associated trend significantly evolves over time.

For both statistical tests, the significance threshold is set at p-value $\leq 0.05$.

A third set of analysis concerns hydrogeological extreme events, defined here as as decadal events, i.e., with a probability to occur each year of 0.1 in present day, for both wet and dry events. The spatial distribution of median time proportion corresponding to these extreme wet events is analysed, per future horizon and per RCP.

All the analyses performed on the total set of available CPs (Table 1) are also individually performed on the four storylines selected in the EXPLORE2 project (see section 2.1) in order to compare trends extracted from all the combined CPs and the ones obtained when the physical consistency of each storyline is conserved.





## 3 Results

### 3.1 Spatial evolution of the french groundwater level over the 21$^{st}$ century

The SPLI maps for each RCP8.5 projection of the median value over the long-term horizon (Figure 3) show heterogeneity between the different projections from dry to wet values. Some GCM/RCM couples (e.g., EC-EARTH/HadREM3-GA7-05) induce negative SPLI values corresponding to dry decadal events (i.e., SPLI below -1.28 more than half of the time) throughout almost all the AquiFR domain, while others induce positive SPLI values corresponding to wet decadal events (e.g., IPSL-CM5A-MR/RCA4). More GCM/RCM couples induce an increase of the groundwater level over the domain (11 out of 17, ≈ 65%), while 4 induce a decrease (≈ 23%) and 2 induce rather stable values (≈ 12%).

Similar analyses were performed for SPLI spatial distribution for all RCP/horizon couples and are available in supplementary material under the RCP2.6 (Appendices A1, A2 and A3) and RCP4.5 (Appendices A4, A5 and A6) respectively on the three horizons, and under the RCP8.5 on the short-term horizon (Appendix A7) and the intermediate horizon (Appendix A8). Despite the divergent results, two main patterns are distinguishable under the RCP8.5:

- from the northern part of the AquiFR domain to the South of the Loire basin, the median of the groundwater levels increases up to what is today considered as a 5-year wet events on the short-term and intermediate horizons, and to wet decadal event on the long-term horizon;

- on the contrary, the median of the groundwater levels remains stable in the southern AquiFR domain (e.g., Tarn-et-Garonne) for the short-term horizon and decrease then to reach a 5-year dry events in the long term horizon.

Projections using RCP2.6 present even larger heterogeneities. On the long-term horizon, 4 out of 10 GCM/RCM couples (40%) project an increase of the groundwater level to a relatively high value, 3 couples (30%) project almost no change and 3 couples (30%) project a decrease of the groundwater, especially the ones related to the EC-EARTH GCM. These results are similar on the three horizons.

Similarly, projections with RCP4.5 for the short-term and intermediate horizons are divergent, preventing any statement as to the future groundwater trend for this scenario. On the long-term horizon, the results are more homogeneous with 6 out of 9 GCM/RCM couples inducing increasing groundwater level corresponding to high values over more than half of the AquiFR domain, while 3 couples induce either no change or slight decrease of the groundwater level.

The results specified to the storylines on the spatial distribution of SPLI (see Figure 4) show contrasted future evolution depending on the storyline. For the orange and green storylines, the trends indicate a strong decrease and increase in groundwater levels respectively, which are fairly continuous until 2100. The orange storyline leads to a increase in the area dedicated to dry events, specific 10-year events which cover almost the entire AquiFR domain on the long-term horizon. On the contrary, the green storyline leads to predominantly stable or increasing groundwater levels for the three horizons. A large area is characterised by median groundwater level above the 10-year wet level on the full period (see Figure 4).

The results of the purple and yellow storylines are more contrasted in space and time. The purple storyline leads to stable groundwater levels on the northern part of the domain for the short-term horizon while in the Beauce and in the Loire basin,





**Figure 3.** Spatial distribution of the SPLI median over the AquiFR domain under the **RCP8.5** on the **long-term horizon (2071-2099)**.



groundwater levels increase (median above 2.5-year wet). This pattern changes over time, with conditions closed to normal for the intermediate and long-term horizon over the major part of the domain, except for the Basse-Normandie region with decreasing trend. The yellow storyline presents the smallest changes. Groundwater levels over the southern Beauce, the Loiret

department and the northern part of the Loire basin slightly decrease (median SPLI below the 2.5-year dry, and around normal elsewhere). Similar results are observed for the long-term horizon, excepted in the northern France (Nord-Pas-de-Calais and Basse-Normandie), where groundwater levels slightly increase (median SPLI between 2.5- and 5-year wet).

Let us note that the MARTHE Tarn-et-Garonne and MARTHE Alsace models have an opposite trend with decreasing ground-water levels for all the storylines, corresponding to decadal dry events, from the short-term horizon for Alsace and from the

long-term horizon for Tarn-et-Garonne.

**Figure 4.** Spatial distribution of the SPLI median for the four storylines over the AquiFR domain under the **RCP8.5** on the three future horizons.





## 3.2 Evolution of the occurrence of wet and dry events

Figure 5 shows the temporal evolution of the percentage of the surface within the AquiFR domain that falls into one of the six SPLI categories depicted in Table 4. The multi-model median, as well as the minimum and maximum, are plotted. The left column corresponds to wet events and the right column to dry events, with the associated envelopes between the minimum and maximum values. The p-values corresponding to the statistical tests and the medians of surface per future horizon dedicated to each category of event are gathered in Table 4. Results show that, for the decadal events (Figure 5, $1^{st}$ line), the envelopes between the minimum and maximum values are wide, highlighting the heterogeneity of the projections especially for decadal events. However, the running median of wet decadal events are projected to spread with time and increasing greenhouse gaz emission from 3.7% of the domain on the reference period to 13.5%, 16.3% and 16.7% respectively for RCP2.6, RCP4.5 and RCP8.5 on the long-term horizon. This is confirmed by the statistical tests with p-values below 0.05 for almost all the 6 categories of events, meaning that the future is significantly different from the reference period (Kolmogorov-Smirnov test) and changes significantly over the $21^{st}$ century (Mann-Kendall test). The Dry decadal events area is also projected to increase, but with a much less pronounced trend, reaching at most 10.0% under the RCP8.5, compare to the $\approx 2.3\%$ in the reference period. For both extremes, the maximum expansion can reach 80% of the area. The only events showing p-values above 0.05 are the 5-year wet events under the RCP8.5 and the 10-year dry event under the RCP4.5, both for the Mann-Kendall test, which indicates no significant future evolution of the dedicated surface proportion.

The changes of surface proportions corresponding to the 5-year dry or wet events are low (Figure 5, $2^{nd}$ line). The surface proportions for 5-year wet events slightly increase from 7.8% on the reference period to 10.0% under the RCP2.6, 10.2% under the RCP4.5 and 9.0% under the RCP8.5 on the long-term horizon. On the contrary, the surface proportions for 5-year dry events slightly decrease from 7.0% on the historical period to 4.3% under RCP2.6 and 4.3% under the RCP4.5 and 5.5% under the RCP8.5 on the long-term horizon.

The changes observed on the surface proportions related to 2.5-year events are the only ones for which the corresponding surface proportions decrease for both wet and dry events (Figure 5, $3^{rd}$ line). Regarding 2.5-year wet events, the evolution is low with the proportion decreasing from 19.6% on the reference period to 18.9% under the RCP2.6, 18.5% under the RCP4.5 and 14.1% under the RCP8.5 on the long-term horizon. Regarding 2.5-year dry events, the decrease is larger, from 18.8% on the reference period to 12.6% with the RCP2.6, 12.0% with the RCP4.5 and 10.5% with the RCP8.5 on the long-term horizon.
 Results from the storylines (see appendix B1 in supplementary material) , are as expected, contrasted. The orange storyline tends to decrease the surface affected by wet events and increase the ones affected by dry events, while the green storyline does the opposite. The changes induced by the purple and the yellow storylines are smaller, excepted for both wet and dry decadal events for which the contrasts are particularly pronounced. The purple storyline induces a noticeable increase on both dry and wet decadal events.





**Figure 5.** Temporal evolution of the 5-year running median of proportion of area from the AquiFR domain affected by the categories of events from Table 4. From top to bottom are represented the 3 return periods, and from left to dry the wet and dry events. The minimum and maximum values over the set of GCM/RCM couples are also plotted. The gap between the historical curve (in grey) and the future curves under the three RCPs corresponds to the exclusion of the 2003-2007 period centred on 2005 in the calculation of running medians (see section 2.3). The dashed black line represents the median value over the historical period gathering all the available projections.





**Table 5.** Results from Kolmogorov-Smirnov (KS) test and Mann-Kendall (MK) test for all the available climate projections on each category of events from Table 4. The direction of the trend is provided if both tests are significant, as well as the corresponding median areas affected for all three horizons.

| | | Historical | RCP2.6 | | | | RCP4.5 | | | | RCP8.5 | | | |
|---|---|---|---|---|---|---|---|---|---|---|---|---|---|---|
| Temporal periods | | 1976 - 2005 | 2021 - 2050 | 2041 - 2070 | 2021 - 2050 | Future general trend | 2021 - 2050 | 2041 - 2070 | 2021 - 2050 | Future general trend | 2021 - 2050 | 2041 - 2070 | 2021 - 2050 | Future general trend |
| ≥ 10-year wet | KS p.value | - | | < 0.001 | | ↗ | | < 0.001 | | ↗ | | < 0.001 | | ↗ |
| | MK p.value | - | | < 0.001 | | | | < 0.001 | | | | < 0.001 | | |
| | AquiFR coverage (%) | 3.7 | 7.7 | 7.2 | 13.5 | | 10.1 | 9.8 | 16.3 | | 9.6 | 12.8 | 16.7 | |
| ≥ 5-year wet | KS p.value | - | | < 0.001 | | ↗ | | < 0.001 | | ↗ | | < 0.001 | | = |
| | MK p.value | - | | < 0.001 | | | | < 0.001 | | | | 0.660 | | |
| | AquiFR coverage (%) | 7.8 | 8.2 | 8.4 | 10.0 | | 9.2 | 8.9 | 10.2 | | 9.0 | 9.7 | 9.0 | |
| ≥ 2.5-year wet | KS p.value | - | | < 0.001 | | = | | < 0.001 | | = | | < 0.001 | | ↘ |
| | MK p.value | - | | 0.001 | | | | 0.004 | | | | < 0.001 | | |
| | AquiFR coverage (%) | 19.6 | 19.1 | 18.1 | 18.9 | | 19.6 | 17.7 | 18.5 | | 18.3 | 17.4 | 14.1 | |
| ≥ 2.5-year dry | KS p.value | - | | < 0.001 | | ↘ | | < 0.001 | | ↘ | | < 0.001 | | ↘ |
| | MK p.value | - | | < 0.001 | | | | < 0.001 | | | | < 0.001 | | |
| | AquiFR coverage (%) | 18.8 | 15.4 | 15.0 | 12.6 | | 15.0 | 15.1 | 12.0 | | 15.1 | 14.0 | 10.5 | |
| ≥ 5-year dry | KS p.value | - | | < 0.001 | | ↘ | | < 0.001 | | ↘ | | < 0.001 | | ↘ |
| | MK p.value | - | | < 0.001 | | | | 0.011 | | | | < 0.001 | | |
| | AquiFR coverage (%) | 7.0 | 6.2 | 6.9 | 4.3 | | 5.3 | 6.3 | 4.3 | | 6.6 | 6.5 | 5.5 | |
| ≥ 10-year dry | KS p.value | - | | < 0.001 | | ↗ | | < 0.001 | | = | | < 0.001 | | ↗ |
| | MK p.value | - | | < 0.001 | | | | 0.603 | | | | < 0.001 | | |
| | AquiFR coverage (%) | 2.3 | 5.0 | 6.2 | 3.7 | | 3.1 | 4.3 | 3.3 | | 6.1 | 6.2 | 10.0 | |

## 3.3 Evolution of the time spent in extreme events

Figure 6 the maps of the multi-model median of the time proportion corresponding to extreme events (i.e., with at least a decadal return period), for both wet (Figure 6, a)) and dry (Figure 6, b)) events. Again, it should be remind that there are not the same number of projections per RCP (see Table 1). The results show that on the short-term horizon, the median of the time proportion corresponding to decadal wet ranges from 25% to 50% over half of the AquiFR domain for the three RCPs, the other part being mostly characterized by values ranging from 10% to 25% (i.e., closer to historical conditions). Then, the median of the time proportion increases along time, reaching from 50% to 75% on RCP4.5 and RCP8.5 on the long-term horizon, located mostly on the northern part of the AquiFR domain.

Results for the time proportion corresponding to dry decadal events are more contrasted. On the short-term horizon on RCP2.6, the median ranges from 10% to 25% on more than half of the AquiFR domain, while the median mostly ranges from 5% to 10% on RCP4.5 and RCP8.5 corresponding to a slight decrease compared to the historical period. However, the tendencies change along time. On the intermediate-term horizon, the values are spatially contrasted for RCP2.6 and RCP8.5, ranging from 5% to 50%, while the values range from 25% to 50% on the large majority of the AquiFR domain on RCP4.5. Eventually, the results for the long-term horizon suggest a decrease of the median of the time proportion in dry decadal event with values ranging from 5% to 10% on more than half of the AquiFR domain on RCP2.6 and RCP4.5. The results on RCP8.5 show a decrease of the time proportion in the northern part of the AquiFR domain (Nord Pas-de-Calais, northern part of the Seine basin) with median values ranging from 5% to 10%, and higher values on the southern part of the domain (Poitou-Charente,





Loire, Basse-Normandie, and Tarn-et-Garonne) and in Alsace, with values mostly ranging from 25% to 50% and some locally reaching 75%. This suggests a North-South contrast in the evolution of the time proportion corresponding to dry decadal events. The results from storylines are consistent with the previous analyses (see appendix B2). The orange storyline tends to reduce the time proportion corresponding to wet decadal event to less than 5% over almost all the AquiFR domain for the three future horizons while it tends to increase the time proportion of dry decadal events to more than 75% over the entire domain. On

the contrary, the green storyline tends to strongly increase the time proportion of wet decadal events while reducing the one corresponding to dry decal events. The purple storyline induces an increase of both wet and dry decacal events, higher as time goes to 2100, over half of the AquiFR domain. The results for the yellow storyline are more spatially and temporally contrasted and do not highlight a noticeable future tendency over the $21^{st}$ century.

## 4   Discussion

### 4.1   Divergent projections limiting the extraction of clear future tendency of groundwater evolution over the $21^{st}$ century

In this study, the evolution of the groundwater is simulated by the AquiFR modelling platform using 36 climatic projections spread over 3 socio-economical scenarios RCPs. Results are analysed based on the Standardised Piezometric Level Index (SPLI), using the 1975-2004 period as reference. The results show that the spatial distribution of the SPLI median differs from

one projection to another, leading to an uncertainty that limits the extraction of a clear trend of future groundwater evolution. This uncertainty is not really surprising as the study area is included into the transition zone of the Atlantic region, where expected trends change from being wetter in the North to drier in the South (Goubanova and Li, 2007; Meaurio et al., 2017). Therefore, the climate projections do not agree on the location of the transition zone, inducing the obtained divergence on the evolution of groundwater levels.

Furthermore, this uncertainty, which is often inherent to hydroclimatic modelling chains based on a multi-model approach, is a recurring point in the literature (Knutti et al., 2013) and requires a proper evaluation (Northrop and Chandler, 2014; Evin et al., 2019). To be accurate, a rigorous uncertainty analysis must consider the contribution of each element in the modelling chain to the total uncertainty: (1) climate scenarios such as RCPs or, more recently, SSPs (Shared Socioeconomic Pathways) in the CMIP6 global exercise (IPCC, 2022); (2) global (GCMs) and regional (RCMs) climate models; (3) debiasing and downscaling

methods such as ADAMONT (Evin et al., 2019) or CDF-t (Michelangeli et al., 2009); (4) hydrological component (models and associated parameters); (5) irreducible fraction of uncertainty due to internal climate variability (Hawkins and Sutton, 2009). As part of the EXPLORE2 project, an evaluation of uncertainty propagation was carried out by Evin et al. (2024) on surface hydrology over France, by estimating the variance inherent in each link in the modelling chain. Their results showed that the main contributor to the total uncertainty is internal climate variability (> 40%). Then, the following sources contributing the

most are the GCMs and the RCMs (> 40% of combined contribution), in similar proportion, especially in North and East of France corresponding to the large majority of the AquiFR domain. These elements are in agreement with the literature, showing that the cumulative GCM/RCM contribution often reaches 40% to 80% of the total uncertainty (Habets et al., 2013;





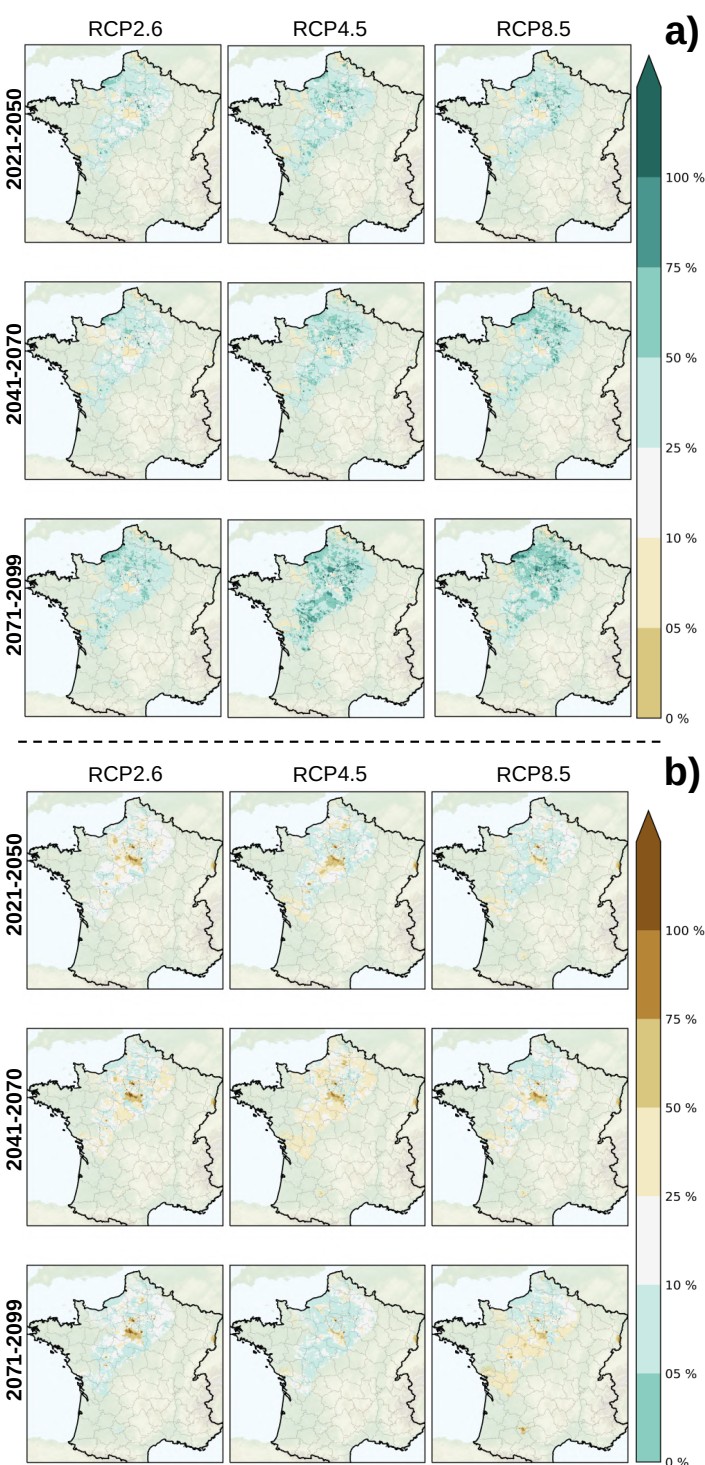

**Figure 6.** Spatial distribution of the median of the time proportion corresponding to extreme wet events (SPLI > +1.28, a)) and extreme dry events (SPLI < -1.28, b)) per future horizon (in line) and per RCP (in column).



Vetter et al., 2015; Tramblay and Somot, 2018; Lemaitre-Basset et al., 2021; Jeantet et al., 2023), and they are consistent with
the analyses showed in this paper suggesting that the results strongly depends on the GCM/RCM couples. In South of France
where the MARTHE Tarn-et-Garonne is included, Evin et al. (2024) showed that the RCPs are the second source contributing
the most to the total uncertainty, which can be related to a weaker influence of precipitation in these regions than on the rest of
the country, and as a consequence a stronger influence of temperature evolution, over the $21^{st}$ .

Despite rigorously assessing the bias induced by using climate projections to force AquiFR, we can still show it when com-
puting SPLI values over the historical period. Indeed, part of the uncertainty is linked to the fact the climate projection cannot
be fully unbiased. It creates uncertainty, since the projection are not compared to present day observations or reference sim-
ulation, but to present day climate simulation. In order to illustrate how this impact the hydrogeological projection analysis,
we compare the CDFs obtained from CPs with a reference CDF obtained using the reference atmospheric forcings over the
historical period. To be consistent with the bias correction method of CPs in the EXPLORE2 project (Marson et al., 2024), the
SAFRAN historical reanalysis product (Vidal et al., 2010; Le Moigne et al., 2020) is used as the meteorological reference. For
each model, the evaluation is carried out by first calculating a CDF of the model extracted per CP over the historical period.
Second, the spatial mean of CDFs from each CP is calculated with a 95% confidence interval. Third, from the simulations
obtained by forcing AquiFR with SAFRAN on the historical period, a reference CDF is calculated. Fourth, the average CDF
obtained from the CPs is compared to the reference CDF. This evaluation is performed for each of the 12 months and Figure 7
illustrates results for January.

For almost all models, the CDFs from CPs are relatively close and distributed on either side of the reference CDFs, with
average projected CDFs following the trends of the reference CDFs quite well. This indicates that the average distribution of
groundwater level obtained from CPs is similar to the distribution obtained from the SAFRAN historical reanalysis over the
historical period. Similar analyses were carried out on the 12 months of the hydrological year and led to similar results. The
largest differences are observed in March and April, suggesting that the CPs diverge on this period, i.e., the end of the wet pe-
riod and the beginning of the dry period and so the end of the recharge period of water tables. This period is also characterised
by significant inter-annual variability, which is difficult to represent in climate models. Overall, these results suggest that the
construction of SPLI by the AquiFR simulations is not noticeably affected by the use of CPs. Therefore, the discrepancies
between the SPLI values observed on the future horizons are unlikely to be induced by the construction of the CDFs from the
CPs on the historical period, and are more likely due to the divergences from the CPs on the future period, and as so from
the climate components of the modelling chain (GCMs and RCMs), which is in agreement with Evin et al. (2024) on surface
hydrology, as mentioned-above.

However, for a specific model, the CDFs from CPs are never fully merged, suggesting discrepancies directly induced by the
adjustment of the CDFs by CPs. Therefore, if these CDFs are too divergent, this may lead to SPLI values related to different
categories of events from one CP to another (Table 4), in particular due to the presence of threshold effects on the studied
aquifers. This may contribute to the uncertainty associated with the SPLI, and then on both sign (dry/wet) and severity of the
corresponding event.

Let us note that the discrepancies of the CDFs of the CPs in present day attest that the bias correction method is not fully





**Figure 7.** Comparison of CDFs from the reference reanalysis SAFRAN with those from CPs calculated on the historical period (see Table 2), for the month of January. The IDs of each model correspond to the abbreviations from Table 3.





efficient, which is expected. Indeed, even a perfect bias correction method would lead to different results due to the internal variability included in each CP. More robust studies are required, accounting for both the need of an accurate method and the
complexity of hydrogeological modelling limiting the current uncertainty analysis.

## 4.2    Groundwater evolution using all the available climate projections

Despite the divergences between the projections, some tendencies may be extracted from the results. The 10 projections from RCP2.6 are too divergent to give confidences on the evolution of the groundwater, especially on the short-term and the interme-
diate horizons. On the long-term horizon, the results are still very divergent but we should note a slight increase of groundwater levels, with levels locally reaching the 10-year wet events levels from the historical period.

The 9 projections from RCP4.5 are also too divergent on the short-term and intermediate-term horizons to extract a clear future tendency. However, 6 over 9 CPs project an increase of groundwater on the long-term horizon, with values reaching the 5-year to 10-year wet return period of the reference. This result is in line with the statement established by Marson et al. (2024) on
the climate aspect of the EXPLORE2 project, which projects a 3.5% increase in precipitation on the long-term horizon under the RCP4.5 in North of France, where most of the AquiFR domain is located. This additional amount of precipitation could increase water supply and ultimately raise groundwater levels in these areas.

Under the RCP8.5, i.e., scenario leading to the largest greenhouse gases emissions, two main contrasted trends are emerging: (1) increasing groundwater levels into the Seine catchment and the northern part of the Loire catchment with an expected in-
crease of wet events, becoming more pronounced over time, and groundwater level reaching the 10-year groundwater level of the reference period; (2) decreasing groundwater levels but staying higher than the levels of 5-year dry events on the reference period on the South of the AquiFR domain on the long-term horizon.

This North-South contrast in the evolution of groundwater levels under the RCP8.5 appears to be the consequence of the North-South contrast observed in future rainfall projected by climate models used in the EXPLORE2 project (Marson et al.,
2024). In this paper, the limited spatial representation of South of France into the AquiFR platform does not allow us to establish with certainty the existence of this North-South divide. However, our results are in agreement with Vergnes et al. (2023), which predict an increase in groundwater level in the northern part of the French metropolitan area as time goes to 2100, and a decrease in the southernmost sectors of France, using a subset of the RCP8.5 DRIAS projections. Moreover, these conclusions are also in line with the main results obtained as part of the EXPLORE2 project on groundwater recharge (Lanini et al., 2024)
and surface hydrology (Sauquet et al., 2024; Evin et al., 2024) over the 21$^{st}$ century in France.

However, some differences with the conclusions extracted from the literature are to be noticed. First, our conclusions for the northern France are not consistent with the results of the EXPLORE 2070 project which, as mentioned in section 1, suggested a global decrease of both recharge and groundwater level over France (Habets et al., 2013; Carroget et al., 2017). The divergence might come from that CPs from the CMIP5 global project are wetter than the ones from the CMIP3 global project
(Meehl et al., 2007) from which the Explore 2070 project was based on, especially over northern France (Dayon et al., 2018). Hydrological systems are therefore directly affected by this additional amount of water into the water balance between the two




versions of the projections. In this way, the consistency between the present results and the ones from Vergnes et al. (2023) was to be expected as they used the hydrogeological MARTHE model, included in the AquiFR platform, and CPs provided by the DRIAS-2020 database, which is included into the CP database established by EXPLORE2. However, the main difference
is that their study is based on 5 selected regionalized CPs, while in this paper 17 different GCM/RCM combinations have been used. The use of a larger panel of CPs allows us to better integrate the uncertainty associated with the modelling chain, in particular the contribution of the CPs, which constitutes the main originality of this study.

Second, most of studies dealing with surface hydrology state that the seasons will be more contrasted under climate change, e.g., winter flows will be higher while summer will be lower (Sauquet et al., 2024), induced by a seasonal regime change of
rainfall and evapotranspiration balances (Soubeyroux et al., 2021; Marson et al., 2024). However, by assessing the spatial distribution of SPLI for each one of the 12 months of the year (not shown), we found similar results for each month, suggesting no modification in the seasonal regime of groundwater level. This phenomenon may come from the slow temporal dynamics characterizing the aquifers in the study area, particularly those close to the Parisian Basin, constituting a large part of the AquiFR domain. Indeed, these aquifers are subject to variations observed on timescales that are often longer than the hydrological year
and even multi-decadal (Baulon et al., 2022), making it difficult to observe sub-annual variations.

A major limitation of this work is that the groundwater abstractions currently applied to the studied aquifers, thus defined according to current water availability and water demand, are maintained in the future and do not change regardless of groundwater level evolution and groundwater availability to meet future water demand. This assumption might be irrelevant as for some projections groundwater level is supposed to decrease over the $21^{st}$ century, specifically in the southern AquiFR domain.
In such cases, current groundwater abstractions need to be adapted (Stollsteiner, 2012) to future conditions. In further assessments, including adapting groundwater abstraction would be an important way to improve our analysis.

### 4.3   Hydrogeological extreme events under climate change

In this study, hydrogeological extreme events are defined as decadal events, i.e., with a probability to occur each year of 0.1 in
present day, for both wet and dry events. The results show that the surface affected by extreme wet events increase by the end of the $21^{st}$ century as both time goes to 2100 and greenhouse gases are released by the RCP, by 7 times under the RCP8.5 on the long-term horizon, and concern the large majority of the AquiFR domain, especially in the northern part. This evolution is consistent with the increase of the median time proportion of these extreme wet events for the 3 RCPs on the 3 future horizons on the long-term horizon, on more than half of the AquiFR domain. The increase of wet events may have consequences on
surface water leading to an increased risk of flooding, as shown by Tramblay et al. (2024) in northern France, due to the strong influence of groundwater dynamics on the amount of water in rivers and wetlands (Guillaumot et al., 2024).

Conversely, the results also show an increase in extreme dry events, but being much less marked than the one for wet events in the large majority of the AquiFR domain. The areas where this increase is larger are Alsace, Poitou-Charentes and Tarn-et-Garonne on the RCP8.5. Therefore, the meteorological droughts announced in the EXPLORE2 project (Marson et al., 2024)
seem to not noticeably involve hydrogeological droughts in the large majority of the AquiFR domain. This is a noticeable dif-





ference with recent hydrological drought assessments which clearly indicate an intensity of drought events, whether in terms of low flows observed or none flow periods, specifically in South of France (Dayon et al., 2018; Lemaitre-Basset et al., 2021; Tramblay et al., 2024). This result is also in contradiction with the results from the EXPLORE 2070 project which suggested a strong increase of hydrogeological droughts (Stollsteiner, 2012; Habets et al., 2013; Carroget et al., 2017). As mentioned

in section 4.2, the CPs used in the EXPLORE2 project are wetter than the ones used in the EXPLORE 2070 project (Dayon et al., 2018). As such, the ones from EXPLORE2 are likely to induce fewer dry events than in the EXPLORE 2070 project. Furthermore, the aquifers corresponding to the study area are defined by annual or multi-annual temporal dynamics (Baulon et al., 2022) and often involve groundwater-river interactions where the groundwater feeds the river (Guillaumot et al., 2022). Therefore, droughts may be observed at the surface without any direct effect on groundwater.

As mentioned in section 4.1, the CPs do not converge and some CPs induce a decrease of groundwater level, e.g., the EC-EARTH/HadREM3-GA7-05 couple involving an increase of +60% of the surface proportion associated to decadal dry events (see section 4.4). In this case, groundwater recharge may decrease, phenomenon which is already observed nowadays in some locations, e.g., the Champagne region (Sobaga et al., 2024).

**4.4 Groundwater evolution using the storylines**

Faced to the difficulty of extracting trends using all the available CPs from the multi-model ensemble approach, the storyline approach is a complementary method (Baulenas et al., 2023). The purpose is to select CPs based on contrasted possible future trends, while maintaining the physical consistency of each chosen CP (Shepherd et al., 2018; Shepherd, 2019), in terms of considered processes, their time evolution and spatial patterns.

As expected, the results from the four storylines defined in EXPLORE2 for the RCP8.5 show contrasted results, strongly related to the design assumptions and rather continuously up to 2100. Following the strong warming and drying hypothesis, the orange storyline induces a large decrease of groundwater level, with median levels in the future corresponding to 10-year dry events in the historical period. On the contrary, the green storyline, which assumes a significant increase in precipitation, leads to predominantly stable or increasing groundwater on the three horizons while the area corresponding to 10-year dry

groundwater level remains close to the historical value. Unlike the previous storylines, the results of the purple and yellow storylines are more contrasted in space and time. The purple storyline leads to stable groundwater in the northern part of the domain on the short-term horizon and to an increase of groundwater water levels in the Beauce and Loire basins. This pattern changes over time, with conditions close to normal on the intermediate horizon and in the long term, a stability of the groundwater on the major part of the domain, with the exception of Basse-Normandie with decreasing trend. The yellow storyline

presents the smallest changes. The groundwater levels on the southern Beauce, the Loiret and the northern Loire basin slightly decrease. Similar results are observed on the long-term horizon, with the exception of the North (Nord Pas de Calais and Basse-Normandie), where groundwater level slightly increases. Let us note that results from the MARTHE Tarn-et-Garonne and MARTHE Alsace models often show groundwater level which decreases, to levels mostly corresponding to decadal dry events from the historical period, regardless the storylines, on the short-term horizon for Alsace and on the long-term horizon





for Tarn-et-Garonne. Eventually, results from the storylines also pointed the probable North-South contrast on groundwater evolution over France, which is in agreement with the trends extracted from the 17 CPs under the RCP8.5 (see section 4.2. These non-convergent groundwater level trends between the storylines raises the question of how the use of these storylines, despite their differences, can be useful to groundwater decision-makers. The northern part of the AquiFR domain, i.e., from Nord Pas-de-Calais to the South of the Loire basin, is the sector where the contrast between the scenarios is the largest, particularly in

the long-term horizon. In this context, the storylines show contrasted yet possible futures, that can invite the decisions-makers to establish their policy using the less favorable projection, i.e., the "worst case scenario" (or precautionary principle), or using no-regrets measures based on the consequences of climate change on other surface variables such as the different seasonal flows (Sauquet et al., 2024) or the QJXA20 (20-year return period quantile of the annual maximum daily flow) to characterise floods (Tramblay et al., 2024). In the southern AquiFR domain, i.e., Poitou-Charentes and Tarn-et-Garonne, the analyses show

more convergent effects, i.e., a drying up of the resource, the intensity of which varies over time. In this context, the decision-maker will be able to consider this phenomenon as a certainty paying particular attention to droughts.

## 5   Conclusions

In this study, the purpose is to identify the main impacts of climate change on groundwater levels in the French regions covered

by the AquiFR platform: Nord Pas-de-Calais, Basse-Normandie, Parisian Basin, Loire, Poitou-Charentes, Alsace and Tarn-et-Garonne. For this purpose, the hydrogeological modelling platform AquiFR, gathering 11 different models (a model being the combination of a study aquifer with the associated hydrogeological software package), is connected to a hydroclimatic modelling chain using an multi-model ensemble approach, with 36 climate projections under RCP2.6, RCP4.5 and RCP8.5. The future evolution of groundwater level is assessed using the SPLI, a normalized indicator of groundwater level, in order to

compare various types of aquifers at once.

The results show future trends strongly divergent, mainly depending on the GCM/RCM couples, either on the spatial distribution of median SPLI or on areas associated with extreme decadal events, defined as 10-year events, i.e., characterised by return periods of more than 10 years. In some RCP / future period combinations, this discrepancy made it difficult to extract clear future trends of groundwater levels.

For the **RCP2.6**, the results are too divergent to extract clear trends of future groundwater level evolution on the short-term (2021-2050) and intermediate (2041-2070) horizons, some showing an evolution (increase or decrease), other showing no noticeable changes. On the long-term horizon (2071-2099), the majority of the projections induces an increase of groundwater level, mostly in the northern AquiFR domain, i.e., Normandie, Nord Pas-de-Calais, Parisian Basin, reaching median level corresponding to 10-year wet events on the historical period (1975-2004).

For the **RCP4.5**, the results are more convergents but the highlighted changes are small on the short-term and intermediate horizons, and become clearer on the long-term horizon with a distinct upward trend, and a noticeable increase of groundwater levels corresponding to decadal wet events on the historical period, by almost 20% by the end of the $21^{st}$ century. This increase



affects the large majority of the AquiFR domain, except for the most remote areas (Tarn-et-Garonne and Alsace) for which the obtained trends are too divergent from one GCM/RCM couple to another.

For the **RCP8.5**, the divergences are the largest, but trends are emerging in the results, with an increase in groundwater levels (with return periods greater than 5 years) in the northern part of the AquiFR domain (Nord Pas-de-Calais, Parisian Basin, Normandie) and a decrease in the southern part (Poitou-Charentes and Tarn-et-Garonne). This is the RCP with the most pronounced North-South differentiation, with a transition zone on the northern border depending on the GCM/RCM couple. It is also the RCP that produces the largest amplification of both dry and wet extreme groundwater levels. Because of the lim-

ited spatial extent in South of France, this study alone is not sufficient to establish the North-South divide highlighted for the RCP8.5. However, the results are consistent with several similar studies dealing with climate change impacts on French hydrology, especially the joined researches carried out as part of the EXPLORE2 project on groundwater recharge (Lanini et al., 2024) and surface hydrology (Sauquet et al., 2024), which justifies the authors' confidence in this conclusion.

Furthermore, the evaluation of the four storylines were selected in the EXPLORE2 project. These storylines highlight the most

extreme changes, for example under a very dry scenario and a very wet scenario, particularly in the North of the AquiFR domain. The use of these storylines might encourage the decision-makers to follow the precautionary principle and choose the worst case situation to establish a sustainable groundwater management policy.

Several potential ways of improvements to this work are conceivable, such as integrating more models into the AquiFR platform and then increasing its representativeness for the different French hydrogeological contexts, especially in South of France.

Similarly, the use of other hydrogeological modelling software per model could make a significant contribution to the reliability of the study, particularly in terms of accurately assessing the propagation of uncertainties along the modelling chain. Eventually, considering the very close relationship between water policy (including water use) and water availability, some future work should integrate socio-economic considerations into physically-based models, then making water demand related to water availability.


*Code availability.* As part of the AquiFR project led by the OFB ("Office Français pour la Biodiversité" in French for French Office for Biodiversity), the AquiFR modelling platform is not directly hosted on any public repository.

*Data availability.* The climate and hydrogeological projections (water table levels and corresponding SPLI values) are available on the DRIAS-Eau database (**https://drias-eau.fr/**, last access on 25th of October, 2024).

*Author contributions.* AJ, JPV, SM and FH conceptualized the work. AJ and JPV performed the simulations and analyses. AJ drafted the paper. JPV, SM and FH all revised the paper and contributed to its analyses and discussions.



*Competing interests.* The authors declare that they have no conflict of interest.

*Acknowledgements.* The authors want to thank the partners of the EXPLORE2 project, which provided the data regarding the climate projections. Especially, the authors want to thank Eric Sauquet for his relevant comments which made this work achievable. Eventually, the authors want also express their gratitude to the Météo-France Meteorological Agency for providing the data used in this work from the SAFRAN database.




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

**Supplementary material**

**Appendix A**

The figures from Appendix A are related to the spatial repartion of the SPLI median over the AquiFR domain under the
810 RCP2.6 (Appendix A1, Appendix A2 and Appendix A3) and under the RCP4.5 (Appendix A4, Appendix A5 and Appendix A6) respectively on the short-term, intermediate-term and long-term horizons. The similar ones are available under the RCP8.5 (Appendix A7 and Appendix A8) on the short-term and intermediate-term horizons.





**Figure A1.** Spatial distribution of the SPLI median over the AquiFR domain under the **RCP2.6** on the **short-term horizon (2021-2050)**.





**Figure A2.** Spatial distribution of the SPLI median over the AquiFR domain under the **RCP2.6** on the **intermediate-term horizon (2041-2070)**.







**Figure A3.** Spatial distribution of the SPLI median over the AquiFR domain under the **RCP2.6** on the **long-term horizon (2071-2099)**.





**Figure A4.** Spatial distribution of the SPLI median over the AquiFR domain under the **RCP4.5** on the **short-term horizon (2021-2050)**.





**Figure A5.** Spatial distribution of the SPLI median over the AquiFR domain under the **RCP4.5** on the **intermediate-term horizon (2041-2070)**.





**Figure A6.** Spatial distribution of the SPLI median over the AquiFR domain under the **RCP4.5** on the **long-term horizon (2071-2099)**.



**Figure A7.** Spatial distribution of the SPLI median over the AquiFR domain under the **RCP8.5** on the **short-term horizon (2021-2050)**.





**Figure A8.** Spatial distribution of the SPLI median over the AquiFR domain under the **RCP8.5** on the **intermediate-term horizon (2041-2070)**.





## Appendix B

The figures from Appendix B are related to the analyses performed on the four storylines, for the temporal evolution of the running median affected by the categories of events (Appendix B1), and for the time proportion spend in extrem events (Appendix B2).





**Figure B1.** Temporal evolution of the running median of proportion from the AquiFR domain affected by the categories of events from Table 4 for the four storylines. The lines represent the return periods of the events, for the wet events (left) and dry events (right). The gap between the historical curve (in grey) and the future curves under the three RCPs corresponds to the exclusion of the 2003-2007 period centred on 2005 in the calculation of running medians (see section 2.3). The black line represents the historical median of each category.





**Figure B2.** Spatial distribution of median of time proportion corresponding to extreme wet events (SPLI > +1.28, a)) and extreme dry events (SPLI < -1.28, b)) per future horizon (in line) for the four storylines (in column) under the RCP8.5.