# Peer review of "Climate change impacts on groundwater simulated using the AquiFR modelling platform"

_EGUsphere, 2025_

## Author Comment (AC1)

**Reply to anonymous referee 2**

We would like to thank the anonymous referee 2 for his truly relevant comments on the manuscript and we are glad he finds this work may show an interest for the hydrogeological community, and may be worth publishing. We agree with most of the comments and we propose an answer for each specific comment hereafter.

**Major Comments**

Comment:

*1. The paper focuses on the impact of climate change but do not address – except at the very end of the conclusion – the strong link between water availability and water demand. One could expect agricultural and water drinking demands to change with climate change, potentially strongly impacting groundwater level. I feel like this issue should be mentioned earlier in the paper and discussed a bit more.*

> ➢ Reply: the referee is right, one can suppose that water demand is expected to change accordingly to the impacts of climate change on water supply. However, addressing anthropisation impacts of water resource was not one of the objectives of the EXPLORE2 project, of which this work is part. Indeed, the idea was to identify climate change impact and anthropisation impact separately, in order to help users to engage into adaptation measures. This point was mentioned in the manuscript as a major limitation of the study (lines 436 to 441). However, the question was assessed on a recent study by Guillaumot et al. (2024). As requested, the authors propose to clarify in the introduction part that this point will not be addressed in this study.

Comment:

*2. The presentation of the results is hard to follow because of the 36 scenarios used and the 4 storylines. I wonder if treating the 4 storylines – that are specifically chosen to illustrate the potential contrasting futures – alone would not be sufficient to give a clear and synthetic message. Especially because the simulated groundwater levels are very uncertain with the 36 scenarios.*

> ➢ Reply: the referee is right. The authors propose to analyse the results from the storylines in the main text and to provide the analyses on all the CPs in supplementary material. However, the authors think that fully removing the analyses on all the CPs is relevant as both sets of analyses are complementary and should not be dealt with separately. This point of view is shared by the EXPLORE2 project community, to which this work is part. A sentence explaining this choice will be provide in the main text after the exploitation of the results on SPLI median from all the CPs.

Comment:

*3. I feel like treating the RCP 2.6 in the paper is not relevant. Because the current CO2 emissions trajectories make it very unlikely, and because the results for this specific RCP are so uncertain that no conclusion can be drawn. I would remove it for sake of simplicity.*

➢ Reply: the referee is right on the fact that the current $CO_2$ emissions trajectories make RCP2.6 very unlikely and that the results induced by RCP2.6 are too divergent. However, the authors propose to move them to supplementary material, to simplify the main text but to still keep them closed to the study, to highlight that the divergences observed from the results mostly depend on the GCM/RCM couples than on the RCPs, which is a difference between climate variables and hydrological variables (Evin et al., 2024).

Comment:

*4. The way SPLI is computed is not clear enough for me. The way the normal CDFs are transposed in SPLI distribution should be explained.*

➢ Reply: the referee is right, the description of the SPLI was not clear enough. The calculation of the Standardized Piezometric Level Index is similar to the calculation of the Standardized Precipitation Index (SPI) (McKee et al., 1993). The SPLI is an indicator used in the French Monthly Hydrological Survey published each month. Details about its computation are given in Seguin (2015). Considering a piezometric head time series of N years, the steps are the following:
  ▪ Step 1: the monthly mean time series from the reference data is computed;
  ▪ Step 2: constitution of twelve monthly time series (January to December) over the N year period. For each time series of N values, a non-parametric kernel density estimation (KDE) allows estimating the best probability density function (pdf) fitting the reference histograms. As time series of piezometric heads show a big variety of histogram, the use of a KDE to estimate a pdf fitting the reference histogram is preferred;
  ▪ Step 3: For each month from January to December, the adjusted cumulative density function is deduced, and a projection over the standardized normal distribution allows deducing the SPLI.
  This part will be clarified in the main text, section 2.3.1, and Figure 2 will be redrawn to better illustrate the calculation method.

Comment:

*5. On all the maps, I would use a very distinct color for the part of France that were not simulated. Figure 3 shows a lot of untreated scenarios. Consequently, the maps for the treated one are very small and hard to read. I don't know how to improve that but having larger map would be great.*

➢ Reply: the referee is right, Figure 3 was unclear. The authors propose a new version (see Figure 1 below) of this type of figures (Figure 3 and the associated elements in supplementary material) to increase the size of each element while keeping the matrix showing the different impacts from either the GCMs or the RCMs. Moreover, the couples used as storylines will be highlighted in this new version, in order to delete the figure dedicated only to the storylines and to simplify the main text.

[Figure]

*Figure 1: Spatial distribution of the SPLI median over the AquiFR domain under the **RCP8.5** on the long-term horizon (**2071-2099**), following the different GCM (in column) / RCM (in line) combination. The last line corresponds to couples where RCMs are used only once. The colored rectangles indicates the GCM/RCM couples corresponding to the storylines.*

Comment:

*6. Figure 5 is really dense, and should be looked at together with Table 5 to follow the text. It's thus quite challenging to catch all the information's presented in part 3.2. Maybe one more argument to keep only the storylines.*

➢ Reply: as requested and as mentioned above, Figure 5 will be replaced by the same version focused on the storylines, and the analyses from all the CPs will be provide in supplementary material. Table 5 will also be put into supplementary material to avoid redundant elements and clarify the main text.

Comment:

*7. I don't really get to what extend the results presented in Vergnes et al, 2023 are different from the MARTHE part of the submitted paper. I would appreciate if this could be discussed somewhere in the introduction.*

➢ Reply: the MARTHE software package used in Vergnes et al. (2023) is based on a simple configuration, with only one layer describing groundwater and a resolution of 2km x 2km, which is larger than the MARTHE configuration used in this paper. Furthermore, the analyses provided by Vergnes et al. (2023) are based on only 5 CPs while this study used 17 GCM/RCM couples, which bring more information specifically regarding uncertainties from the CPs. This point constitutes the main difference between the two studies, and was mentioned in the paper (lines 422-427). The authors propose to clarify the corresponding section to make this point clearer, and also to mention it earlier in the introduction.

Comment:

*8. In part 2.2.1, it's not very clear whether SURFEX or ISBA or both are used for the study. This part is rather confusing and unclear. I think it should be improved.*

➢ Reply: the referee is right, this part in unclear. ISBA is the land surface scheme dedicated to the interaction between the atmosphere, the soil and the biosphere. It is part of the SURFEX modeling platform which is basically designed to integrate several surface schemes such as ISBA, TEB for cities, SEA for sea (Le Moigne et al., 2020). The authors propose to clarify this point in the main text.

Comment:

*9. Multi-model is advocated in the introduction. One could expect that this approach is used throughout the study. It is clearly done for the climate part but not for the groundwater modelling part. If I get it right, most of the time MARTHE and EauDyssé were not applied on the same region. And it's hard to get where it's been done, as all the surfaces in Table 3 are different. I think the reasons why only one groundwater model has been applied should be stated somewhere. And Table 3 should be improved so that locations where both models are applied appear clearly.*

➢ Reply: the referee is right, this point is unclear. Strictly speaking, we cannot say that multi-model approach is used for hydrogeological modelling. Some applications do share parts of their spatial coverage, e.g., the Somme region, but they are few and their configuration (i.e., their assumptions, spatial resolution and spatial coverage) makes them not directly comparable. Furthermore, multi-model approach has already been analysed in the RexHySS project (Habets et al., 2013), which showed on the Somme and the Seine regions that the elements from the modelling chain bringing most of uncertainty are the climate projections. Then the choice was made not to consider it here, in addition to the technical difficulty of implementing several hydrogeological models for each application of the AquiFR platform. The authors propose to clarify this point in the introduction section, and to add a comment in the discussion (section 4.1) to remind why hydrogeological multi-modelling was not used.

**Minor Comments**

Comment:
*1. First sentence of the introduction is not clear for me. Especially "for both agricultural purposes". Please rephrase.*

➢ Reply: as requested, this sentence will be rephrased.

*2. Line 83: typo – replace dot by coma*

➢ Reply: as requested, this issue will be addressed.

*3. How are the weather regimes defined?*

➢ Reply: the weather regimes are based on the cluster of the geopotential height at 500 hPa of the ERA-40 analysis (Verfaillie et al., 2017), on a seasonal basis: (1) December, January, February (DJF); (2) March, April, May (MAM); (3) June, July, August (JJA); (4) September, October, November (SON). The weather regimes are based on the 500 hPa geopotential height cluster of the ERA-40 analysis. This point will be clarified in the manuscript.

*4. Line 117 : typo (Vergnes et al, …)*

➢ Reply: as requested, this issue will be addressed

*5. Line 128 : typo - to simulate*

➢ Reply: as requested, this issue will be addressed

*6. Line 148: "vertical exchange through aquitards" – not clear for me. Aquitards are not supposed to conduct water.*

➢ Reply: the referee is right, hydraulic conductivity in aquitards can be very low. However, aquitards can also be semi-permeable and characterised by low (but not null) hydraulic conductivity. Consequently, vertical exchange through such aquitards are taken into account in SAM in order to integrate confined aquifers.

*7. The way unsaturated zone is treated in MARTHE should be detailed.*

➢ Reply: the referee is right, the unsaturated zone in MARTHE is not fully described in the applications used, and a lag is simulated in simple places using a reservoir approach. However, the authors think that it is not necessary to go into further detail on this point in order not to make the main text more voluminous, in particular as requested by referee 1. The reader is encouraged to refer himself to Habets et al. (2021) for further details, as mentioned at the line 160.

*8. Line 228: I don't get what is meant by "physical consistency of each storyline is conserved"*

➢ Reply: the sentence was unclear. The focus on storyline makes it possible not to present means and statistical results, but to present few plausible futures with all their physical consistency (Marson et al., 2024). The approach is therefore more deterministic regarding the scenario represented by the storyline. The point will be clarified in the manuscript.

*9. Line 304: typo "Figure 6 the maps". Please rephrase*

➢ Reply: as requested, this issue will be addressed.

**References:**

Evin, G., Reverdy, A., and Hingray, B.: Ensemble de projections Explore2 : changements moyens et incertitudes associés, IGE, INRAE, CNRS, Grenoble, France, 2024.

Guillaumot, L., Munier, S., Abhervé, R., Vergnes, J.-P., Jeantet, A., Le Moigne, P., and Habets, F.: Are regional groundwater models suitable for simulating wetlands, rivers and intermittence? The example of the French AquiFR platform, Journal of Hydrology, 644, 132019, https://doi.org/10.1016/j.jhydrol.2024.132019, 2024.

Habets, F., Boé, J., Déqué, M., Ducharne, A., Gascoin, S., Hachour, A., Martin, E., Pagé, C., Sauquet, E., Terray, L., Thiéry, D., Oudin, L., and Viennot, P.: Impact of climate change on the hydrogeology of two basins in northern France, Climatic Change, 121, 771–785, https://doi.org/10.1007/s10584-013-0934-x, 2013.

Habets, F., Amraoui, N., Thiéry, D., Vergnes, J.-P., Morel, T., Le Moigne, P., Munier, S., Leroux, D., de Dreuzy, J.-R., Longuevergne, L., Ackerer, P., Besson, F., Etchevers, P., Rousset, F., Willemet, J.-M., Viennot, P., and Gallois, N.: Plateforme de modélisation hydrogéologique nationale AQUI-FR, OFB, Paris, France, 2021.

Le Moigne, P., Besson, F., Martin, E., Boé, J., Boone, A., Decharme, B., Etchevers, P., Faroux, S., Habets, F., Lafaysse, M., Leroux, D., and Rousset-Regimbeau, F.: The latest improvements with

SURFEX v8.0 of the Safran–Isba–Modcou hydrometeorological model for France, Geoscientific Model Development, 13, 3925–3946, https://doi.org/10.5194/gmd-13-3925-2020, 2020.

Marson, P., Corre, L., Soubeyroux, J.-M., Sauquet, E., Robin, Y., Vrac, M., and Dubois, C.: Rapport de synthèse sur les projections climatiques régionalisées, METEO FRANCE, INRAE, Institut Pierre-Simon Laplace, Toulouse, France, 2024.

McKee, T. B., Doesken, N. J., and Kleist, J.: The relationship of drought frequency and duration to time scales, Proceedings of the 8th Conference on Applied Climatology, California, 179–183, 1993.

Seguin, J.-J.: Proposition d'un indicateur piézométrique standardisé pour le Bulletin de Situation Hydrologique 'Nappes,' BRGM, Orléans, France, 2015.

Verfaillie, D., Déqué, M., Morin, S., and Lafaysse, M.: The method ADAMONT v1.0 for statistical adjustment of climate projections applicable to energy balance land surface models, Geoscientific Model Development, 10, 4257–4283, https://doi.org/10.5194/gmd-10-4257-2017, 2017.

Vergnes, J.-P., Caballero, Y., and Lanini, S.: Assessing climate change impact on French groundwater resources using a spatially distributed hydrogeological model, Hydrological Sciences Journal, 68, 209–227, https://doi.org/10.1080/02626667.2022.2150553, 2023.

---

## Author Comment (AC2)

**Reply to anonymous referee 1**

We would like to thank the anonymous referee 1 for his truly relevant comments on the manuscript and we are glad he finds this work presents an interest for the hydrogeological community. We agree with most of the comments and we propose an answer for each specific comment hereafter.

Comment:

*1. I believe that the manuscript should be divided into two separate papers. In the paper discussed now, only the most important descriptions, results and discussions of three types of statistical analyses performed for only four storylines should be left. However, storyline No. 4 seems unlikely, so its description and results should be moved to the Supplement Material. Similarly, the RCP2.6 path is rather unlikely, so the description of the study and results for this path can also be moved from the main text to the Supplement Material.*

> ➢ Reply: the referee is right about the manuscript being quite long, which does not help for the clarity of the main messages. However, the authors do not believe that splitting this article into two different articles, a first describing results from all the CPs and a second focusing on the storylines, is relevant as they consider that both sets of analyses are complementary and should not be dealt with separately. This point of view is shared by the EXPLORE2 project community, to which this work is part. However, the authors understand the comment of the referee and propose to mainly focus the analyses on the storylines inside the main text and to provide the analyses from all the CPs in the supplementary material, as the referee suggested. Moreover, the authors propose to improve the readability of the manuscript by reducing some redundant elements to supplementary material.

Comment:

*2. A report of studies based on all the available climate projections (CP) can be successfully published in the second paper, in the discussion of which the obtained results can be related to the results for the storylines. Or vice versa - depending on the authors' decision.*

> ➢ Reply: as mentioned in the response to the first comment of the reviewer, the authors believe that splitting this article to two different articles is not relevant as both sets of analyses are complementary and should not be dealt with separately. The authors propose to focus in the main text on the storylines and to move the analyses from all the CPs to supplementary material. In the main article, the spread on the different CPs will still be first presented on Figure 3, in order to show how relevant are the four storylines.

Comment:

*3. The justification for removing descriptions of tests performed for RCP2.6 from the main text are, among others, section 3.2, Fig. 5 and Table 5. This is all unclear and too far removed from the main thrust of this paper, i.e. spatial projection of climate change on a regional scale. Considering the description in the text, it seems possible to move Table 5 to the Supplement.*

> ➢ Reply: the authors agree with the referee's suggestion. As mentioned in the previous responses, this part will be focus on the storylines and the results from the statistical tests (previously Table 5) will be moved to supplementary material.

Comment:

*4. Table 1 should rather be in the Supplement because it is additional information and does not directly concern the main research problem solved by this study.*

> ➢ Reply: as requested, Table 1 will be moved to supplementary material.

Comment:

*5. line 113 – it is unclear what "strong seasonal contrast" means in story No. 3 (Purple) – this should be explained.*

> ➢ Reply: the sentence was unclear, the seasonal contrast being associated to precipitation. The purple storyline is characterised by "strong seasonal contrast **in precipitation**" with the winter season having large amount of precipitation, while summer season has low amount of precipitation (Marson et al., 2024). This point will be clarify in the main text.

Comment:

*6. lines 140-143 - models EROS and HS1D - since these 2 models have not been taken into account in this study, why mention them (it only makes the text harder to follow).*

> ➢ Reply: as requested, to shorten the text, the mentions and descriptions of the models EROS and HS1D will be removed from the main text.

Comment:

*7. Lines 144 – 160 - models EauDyssee and MARTHE - these descriptions are not necessary in the main text and can be moved to the Supplement. Only the sentence in lines 157-159 is important for the main text.*

> ➢ Reply: the authors are willing to keep a short description of the two hydrogeological software packages in the main is important as this prevents the reader to refer himself to the supplementary material to quickly understand the processes modelled by the AquiFR platform. However, as suggested by the reviewer this part will be reduced and clarified.

Comment:

*8.lines 173 -177 - excessive detail - can be moved to the Supplement.*

> ➢ Reply: as for the description of the hydrogeological software packages, the authors think that moving this part to supplementary material is not necessary as Table 3 will be moved to supplementary material, which will clarify this section without requiring the reader to refer to the supplementary material excessively.

Comment:

*9. Table 3 - can be moved to the Supplement because Figure 3 shows the modelled catchments well.*

> ➢ Reply: as requested and as mentioned as the response to the previous comment, Table 3 will moved to supplementary material.

Comment:

*10. lines 193-196 - I don't understand how the SPLI values were calculated. Are they based on frequency percentages? The description should be clearer, because that's the basic element of this research.*

> ➢ Reply: the referee is right, the description of the SPLI was not clear enough. The calculation of the Standardized Piezometric Level Index is similar to the calculation of the Standardized Precipitation Index (SPI) (McKee et al., 1993). The SPLI is an indicator used in the French Monthly Hydrological Survey published each month. Details about its computation are given in Seguin (2015). Considering a piezometric head time series of N years, the steps are the following:
> • Step 1: the monthly mean time series from the reference data is computed;
> • Step 2: constitution of twelve monthly time series (January to December) over the N year period. For each time series of N values, a non-parametric kernel density estimation (KDE) allows estimating the best probability density function (pdf) fitting the reference histograms. As time series of piezometric heads show a big variety of histogram, the use of a KDE to estimate a pdf fitting the reference histogram is preferred;
> • Step 3: For each month from January to December, the adjusted cumulative density function is deduced, and a projection over the standardized normal distribution allows deducing the SPLI.
> This part will be clarified in the main text, section 2.3.1, and Figure 2 will be redrawn to better illustrate the calculation method.

Comment:

*11. sections 4.1 and 4.2 - confirm that this manuscript is too burdened with too much information, analysis and discussion. All of these strays too far in different directions from the main topic given in the title. This should be an article about just one specific study - either about the storylines or about the all available CPs , depending on the authors' decision. It is also possible to propose two publications under one common main title, divided into Parts 1 and 2, respectively covering the indicated topics.*

> ➢ Reply: the referee is right that the manuscript is too long, but as mentioned in the previous responses, the authors do not believe that splitting this article into two different articles is relevant. They propose to focus the main analyses on the storylines and to

provide the analyses on all the CPs from the three RCPs in supplementary material. Ultimately, the discussion will be simplified, clarified and focused on the long-term horizon, which may be sufficient to address the issue raised by the referee about the lack of key messages in the paper and its length.

**References:**

Marson, P., Corre, L., Soubeyroux, J.-M., Sauquet, E., Robin, Y., Vrac, M., and Dubois, C.: Rapport de synthèse sur les projections climatiques régionalisées, METEO FRANCE, INRAE, Institut Pierre-Simon Laplace, Toulouse, France, 2024.

McKee, T. B., Doesken, N. J., and Kleist, J.: The relationship of drought frequency and duration to time scales, Proceedings of the 8th Conference on Applied Climatology, California, 179–183, 1993.

Seguin, J.-J.: Proposition d'un indicateur piézométrique standardisé pour le Bulletin de Situation Hydrologique 'Nappes,' BRGM, Orléans, France, 2015.